# Decoupling Features in Hierarchical Propagation for Video Object Segmentation

**Zongxin Yang**[1,2]**, Yi Yang**[1†]

[1] CCAI, College of Computer Science and Technology, Zhejiang University  [2] Baidu Research
{yangzongxin, yangyics}@zju.edu.cn

## Abstract

This paper focuses on developing a more effective method of hierarchical propagation for semi-supervised Video Object Segmentation (VOS). Based on vision transformers, the recently-developed Associating Objects with Transformers (AOT) approach introduces hierarchical propagation into VOS and has shown promising results. The hierarchical propagation can gradually propagate information from past frames to the current frame and transfer the current frame feature from object-agnostic to object-specific. However, the increase of object-specific information will inevitably lead to the loss of object-agnostic visual information in deep propagation layers. To solve such a problem and further facilitate the learning of visual embeddings, this paper proposes a Decoupling Features in Hierarchical Propagation (DeAOT) approach. Firstly, DeAOT decouples the hierarchical propagation of object-agnostic and object-specific embeddings by handling them in two independent branches. Secondly, to compensate for the additional computation from dual-branch propagation, we propose an efficient module for constructing hierarchical propagation, *i.e.*, Gated Propagation Module, which is carefully designed with single-head attention. Extensive experiments show that DeAOT significantly outperforms AOT in both accuracy and efficiency. On YouTube-VOS, DeAOT can achieve 86.0% at 22.4fps and 82.0% at 53.4fps. Without test-time augmentations, we achieve new state-of-the-art performance on four benchmarks, *i.e.*, YouTube-VOS (86.2%), DAVIS 2017 (86.2%), DAVIS 2016 (92.9%), and VOT 2020 (0.622). Project page: https://github.com/z-x-yang/AOT.

## 1 Introduction

Video Object Segmentation (VOS), which aims at recognizing and segmenting one or multiple objects of interest in a given video, has attracted much attention as a fundamental task of video understanding. This paper focuses on semi-supervised VOS, which requires algorithms to track and segment objects throughout a video sequence given objects' annotated masks at one or several frames.

Early VOS methods are mainly based on finetuning segmentation networks on the annotated frames [7, 32, 51] or constructing pixel-wise matching maps [10, 50]. Based on the advance of attention mechanisms [5, 48, 53], many attention-based VOS algorithms have been proposed in recent years and achieved significant improvement. STM [34] and the following works [11, 43, 44] leverage a memory network to store and read the target features of predicted past frames and apply a non-local attention mechanism to match the target in the current frame. Furthermore, AOT [61, 63, 65] introduces hierarchical propagation into VOS based on transformers [8, 48] and can associate multiple objects

---

†: the corresponding author.

36th Conference on Neural Information Processing Systems (NeurIPS 2022).

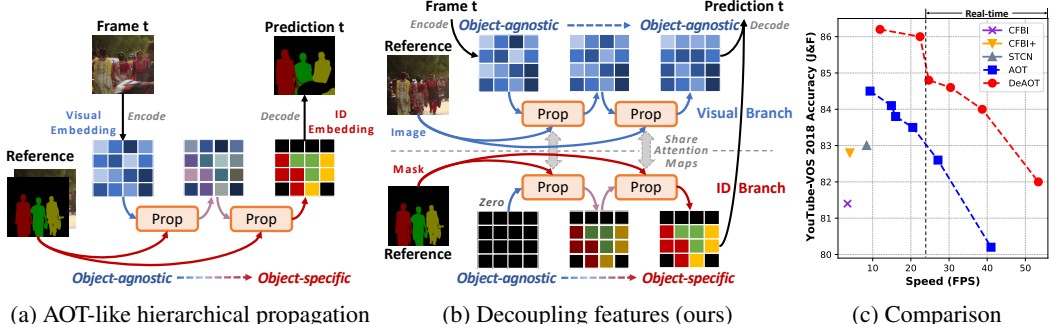

| (a) AOT-like hierarchical propagation | (b) Decoupling features (ours) | (c) Comparison |

Figure 1: (a) AOT [63] hierarchically propagates (Prop) *object-specific* information (*i.e.*, specific to the given object(s)) into the *object-agnostic* visual embedding. (b) By contrast, DeAOT decouples the propagation of visual and ID embeddings in two branches. (c) Speed-accuracy comparison. All the results were fairly recorded on the same device, 1 Tesla V100 GPU.

collaboratively by utilizing the IDentification (ID) mechanism [63]. The hierarchical propagation can gradually propagate ID information from past frames to the current frame and has shown promising VOS performance with remarkable scalability.

Fig. 1a shows that AOT's hierarchical propagation can transfer the current frame feature from an object-agnostic visual embedding to an object-specific ID embedding by hierarchically propagating the reference information into the current frame. The hierarchical structure enables AOT to be structurally scalable between state-of-the-art performance and real-time efficiency. Intuitively, the increase of ID information will inevitably lead to the loss of initial visual information since the dimension of features is limited. However, matching objects' visual features, the only clues provided by the current frame, is crucial for attention-based VOS solutions. To avoid the loss of visual information in deeper propagation layers and facilitate the learning of visual embeddings, a desirable manner (Fig. 1b) is to decouple object-agnostic and object-specific embeddings in the propagation.

Based on the above motivation, this paper proposes a novel hierarchical propagation approach for VOS, *i.e.*, Decoupling Features in Hierarchical Propagation (DeAOT). Unlike AOT, which shares the embedding space for visual (object-agnostic) and ID (object-specific) embeddings, DeAOT decouples them into different branches using individual propagation processes while sharing their attention maps. To compensate for the additional computation from the dual-branch propagation, we propose a more efficient module for constructing hierarchical propagation, *i.e.*, Gated Propagation Module (GPM). By carefully designing GPM for VOS, we are able to use single-head attention to match objects and propagate information instead of the stronger multi-head attention [48], which we found to be an efficiency bottleneck of AOT [63].

To evaluate the proposed DeAOT approach, a series of experiments are conducted on three VOS benchmarks (YouTube-VOS [57], DAVIS 2017 [39], and DAVIS 2016 [38]) and one Visual Object Tracking (VOT) benchmark (VOT 2020 [24]). On the large-scale VOS benchmark, YouTube-VOS, the DeAOT variant networks remarkably outperform AOT counterparts in both accuracy and run-time speed as shown in Fig. 1c. Particularly, our R50-DeAOT-L can achieve **86.0%** at a nearly real-time speed, **22.4fps**, and our DeAOT-T can achieve **82.0%** at **53.4fps**, which is superior compared to AOT-T [63] (80.2%, 41.0fps). Without any test-time augmentations, our SwinB-DeAOT-L achieves top-ranked performance on four VOS/VOT benchmarks, *i.e.*, YouTube-VOS 2018/2019 (**86.2%/86.1%**), DAVIS 2017 Val/Test (**86.2%/82.8%**), DAVIS 2016 (**92.9%**), and VOT 2020 (**0.622 EAO**).

Overall, our contributions are summarized below:

- We propose a highly-effective VOS framework, DeAOT, by decoupling object-agnostic and object-specific features in hierarchical propagation. DeAOT achieves top-ranked performance and efficiency on four VOS/VOT benchmarks [24, 38, 39, 57].

- We design an efficient module, GPM, for constructing hierarchical matching and propagation. By using GPM, DeAOT variants are consistently faster than AOT counterparts, although DeAOT's propagation processes are twice as AOT's.

## 2 Related Work

**Semi-supervised Video Object Segmentation.** Given a video with one or several annotated frames (the first frame in general), semi-supervised VOS [52] requires algorithms to propagate the mask annotations to the entire video. Traditional methods often solve an optimization problem with an energy defined over a graph structure [2, 4, 49]. Based on deep neural networks (DNN), deep learning based VOS methods have achieved significant progress and dominated the field in recent years.

*Finetuning-based Methods.* Early DNN-based methods rely on fine-tuning pre-trained segmentation networks at test time to make the networks focus on the given object. Among them, OSVOS [7] and MoNet [56] propose to fine-tune pre-trained networks on the first-frame annotation. OnAVOS [51] extends the first-frame fine-tuning by introducing an online adaptation mechanism. Following these approaches, MaskTrack [37] and PReM [32] further utilize optical flow to help propagate the segmentation mask from one frame to the next.

*Template-based Methods.* To avoid using the test-time fine-tuning, many researchers regard the annotated frames as templates and investigate how to match with them. For example, OSMN [60] employs a network to extract object embedding and another one to predict segmentation based on the embedding. PML [10] learns pixel-wise embedding with the nearest neighbor classifier, and VideoMatch [22] uses a matching layer to map the pixels of the current frame to the annotated frame in a learned embedding space. Following these methods, FEELVOS [50] and CFBI(+) [62, 64] extend the pixel-level matching mechanism by additionally doing local matching with the previous frame, and RPCM [58] proposes a correction module to improve the reliability of pixel-level matching. Instead of using matching mechanisms, LWL [6] proposes to use an online few-shot learner to learn to decode object segmentation.

*Attention-based Methods.* Based on the advance of attention mechanisms [5, 48, 53], STM [34] and the following works (*e.g.*, KMN [43] and STCN [11]) leverage a memory network to embed past-frame predictions into memory and apply a non-local attention mechanism on the memory to propagate mask information to the current frame. Differently, SST [17] proposes to calculate pixel-level matching maps based on the attention maps of transformer blocks [48]. Recently, AOT [61, 63, 65] introduces hierarchical propagation into VOS and can associate multiple objects collaboratively with the proposed ID mechanism.

**Visual Transformers.** Transformers [48] was initially proposed to build hierarchical attention-based networks for natural language processing (NLP). Compared to RNNs, transformer networks model global correlation or attention in parallel, leading to better memory efficiency, and thus have been widely used in NLP tasks [15, 40, 46]. Similar to Non-local Neural Networks [53], transformer blocks compute correlation with all the input elements and aggregate their information by using attention mechanisms [5]. Recently, transformer blocks were introduced to computer vision and have shown promising performance in many tasks, such as image classification [16, 30, 47], object detection [8]/segmentation [25, 35, 54, 66], image generation [36], and video understanding [1, 26, 31].

Based on transformers, AOT [63] proposes a Long Short-Term Transformer (LSTT) structure for constructing hierarchical propagation. By hierarchically propagating object information, AOT variants [63] have shown promising performance with remarkable scalability. Unlike AOT, which shares the embedding space for object-agnostic and object-specific embeddings, we propose to decouple them into different branches using individual propagation processes. Such a dual-branch paradigm avoids the loss of object-agnostic information and achieves significant improvement. Besides, a more efficient structure, GPM, is proposed for hierarchical propagation.

## 3 Rethinking Hierarchical Propagation for VOS

Attention-based VOS methods [11, 34, 43, 63] are dominating the field of VOS. In these methods, STM [34] and following algorithms [11, 43] uses a single attention layer to propagate mask information from memorized frames to the current frame. The use of only a single attention layer restricts the scalability of algorithms. Hence, AOT [63] introduces hierarchical propagation into VOS by proposing the Long Short-term Transformer (LSTT) structure, which can propagate the mask information in a hierarchical coarse-to-fine manner. By adjusting the layer number of LSTT, AOT variants can be ranged from state-of-the-art performance to real-time run-time speed.

Let $Q \in \mathbb{R}^{HW \times C}$ and $K, V \in \mathbb{R}^{THW \times C}$ denote the query embedding of the current frame, the key embedding, and the value embedding of the memorized frames respectively, where $T$, $H$, $W$, $C$ represent the temporal, height, width, and channel dimensions. The formula of a common attention-based VOS propagation is,

$$Att(Q, K, V) = Corr(Q, K)V = softmax(\frac{QK^{tr}}{\sqrt{C}})V, \qquad (1)$$

where the matching (or attention) map is calculated by the correlation function, $Corr(*, *)$.

To formulate a hierarchical propagation with $L$ layers, we further define $X_l^t \in \mathbb{R}^{HW \times C}$ as the input feature embedding of $l$-th propagation layer ($l \in \{1, 2, ..., L\}$) at $t$ frame. Moreover, $X_l^{\mathbf{m}} = Concat(X_l^{m_1}, ..., X_l^{m_T})$ and $Y^{\mathbf{m}} = Concat(Y^{m_1}, ..., Y^{m_T})$ stands for the feature embeddings and object masks in the memorized frames with indices $\mathbf{m} = \{m_1, ..., m_T\}$. Then, the formulation of $l$-th propagation layer in AOT's hierarchical propagation can be simplified as,

$$\widetilde{X}_l^t = Att(X_l^t W_l^K, X_l^{\mathbf{m}} W_l^K, X_l^{\mathbf{m}} W_l^V + ID(Y^{\mathbf{m}})), \qquad (2)$$

where $ID(*)$ denotes the IDentification (ID) embedding [63] function used to encode masks. Besides, $W_l^K \in \mathbb{R}^{C \times C_k}$ and $W_l^V \in \mathbb{R}^{C \times C_v}$ are trainable parameters for projecting features into matching space and propagation space, respectively. For simplicity, the formulation keeps only the parts related to mask propagation in LSTT.

Obviously, before all the propagation layers, the current frame feature, $X_1^t$, is an object-agnostic feature extracted from an image encoder (*e.g.*, ResNet-50 [21]). Nevertheless, the mask information $ID(Y^{\mathbf{m}})$ will be gradually and hierarchically propagated into the current frame, and the output feature, $\widetilde{X}_L^t$, will become object-specific and can be decoded into the ID/mask prediction by a decoder network (*e.g.*, FPN [27]). In other words, step by step, the hierarchical propagation transfers the current frame feature, $X_l^t$, from an object-agnostic visual embedding to an object-specific ID embedding, as demonstrated in Fig. 1a.

Intuitively, the absorption of object-specific ID information will inevitably lead to the oblivion of object-agnostic visual information within $X_1^t$ since the channel dimension of $X_l^t$ is limited. Such a phenomenon can also be observed by increasing the ID information directly. As shown in Fig. 2, the performance of AOT heavily drops as we increase the information amount of $ID(Y^{\mathbf{m}})$ by containing more IDs inside. On the other hand, the significant progress of VOS in recent years is mainly based on matching object-agnostic visual embeddings (*e.g.*, pixel-level matching methods [58,62,64] and single-layer attention-based methods [11,34,43] mentioned above). Hence, we argue that the loss of visual information in deeper propagation layers limits the performance of hierarchical propagation.

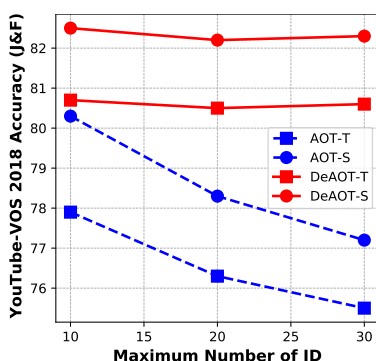

Figure 2: The performance of AOT [63] will be degraded by increasing ID's maximum number.

*How to design a hierarchical propagation structure which can keep or even refine the initial object-agnostic visual information?* Fig. 1b shows a simple, straightforward, and desirable approach, *i.e.*, propagating object-agnostic and object-specific information in two different branches (Visual Branch and ID Branch). The object-agnostic branch is responsible for gathering visual information, refining visual features, and matching objects. By contrast, the object-specific branch is responsible for absorbing ID information propagated from memorized frames. These two branches share the attention maps used to match objects and propagate features. Compared to the single-branch LSTT, our dual-branch approach can keep and further refine visual features in the hierarchical propagation and thus can further facilitate the learning of visual embeddings.

## 4 Decoupling Features in Hierarchical Propagation

This section will introduce a new framework, Decoupling Features in Hierarchical Propagation (DeAOT), for solving semi-supervised video object segmentation. We show an overview of DeAOT in Fig. 3a. Given a video with a reference frame annotation, DeAOT propagates the annotation to the entire video frame-by-frame. The multi-object annotation is encoded by the IDentification

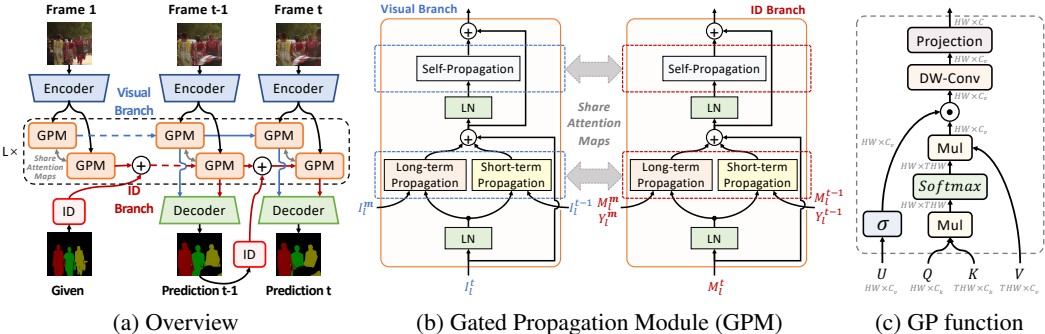

| (a) Overview | (b) Gated Propagation Module (GPM) | (c) GP function |

Figure 3: (a) Overview. Decoupling Features in Hierarchical Propagation (DeAOT) decouples the propagation of visual embedding and IDentification (ID) embedding [63] in two branches, *i.e.*, Visual Branch and ID Branch. The propagation module is the proposed efficient GPM module. (b) A demonstration of the Gated Propagation Module (GPM) in both Visual and ID branches. LN: Layer Normalization [3]. (c) We propose to use the Gated Propagation (GP) function to construct GPM. DW-Conv: depth-wise convolution. Mul: matrix multiplication.

(ID) mechanism [63]. Different from AOT, DeAOT decouples the hierarchical propagation of visual embedding and ID embedding, *i.e.*, DeAOT propagates these two embeddings in two branches. Furthermore, DeAOT constructs the hierarchical propagation by using the proposed Gated Propagation Module (GPM), which is more efficient and effective than the LSTT block used in AOT.

## 4.1 Hierarchical Dual-branch Propagation

Different from the previous attention-based VOS methods [34, 43, 44, 63], DeAOT propagates objects' visual features and mask features in two parallel branches. In detail, the visual branch is responsible for matching objects, gathering past visual information, and refining object features. To re-identify the objects, the ID branch reuses the matching maps (attention maps) calculated by the visual branch to propagate the ID embedding (encoded by the ID mechanism [63]) from past frames to the current frame. Both the branches share the same hierarchical structure with $L$ propagation layers.

**Visual Branch** is responsible for matching objects by calculating attention maps on patch-wise visual embeddings. The visual embeddings in the memorized frames will be propagated to the current frame regarding the attention maps. Since the propagation is not directly related to the object-specific ID embedding, the visual branch can learn to refine visual embeddings to be more contrastive but avoid being biased toward the given object-specific information. Let $I$ denote visual embeddings, we modify Eq. 2 into a layer of object-agnostic visual propagation,

$$
\begin{aligned}
\widetilde{I}_l^t &= Att(I_l^t W_l^K, I_l^{\mathbf{m}} W_l^K, I_l^{\mathbf{m}} W_l^V) \\
&= Corr(I_l^t W_l^K, I_l^{\mathbf{m}} W_l^K) I_l^{\mathbf{m}} W_l^I,
\end{aligned}
\tag{3}
$$

which doesn't leverage the object-specific ID embedding, $ID(Y^{\mathbf{m}})$. Thus, the visual branch can learn to keep and refine the visual embedding in the hierarchical propagation.

**ID Branch** is designed for propagating the object-specific information from past frames to the current frame. The prediction of object-specific segmentation is essential for VOS and can not be processed by the above object-agnostic visual propagation branch. Let $M$ denote the object-specific embeddings in our identification branch, the formulation of our object-specific ID propagation is,

$$
\begin{aligned}
\widetilde{M}_l^t &= Att(I_l^t W_l^K, I_l^{\mathbf{m}} W_l^K, M_l^{\mathbf{m}} W_l^{\overline{V}} + ID(Y^{\mathbf{m}})) \\
&= Corr(I_l^t W_l^K, I_l^{\mathbf{m}} W_l^K)(M_l^{\mathbf{m}} W_l^{\overline{V}} + ID(Y^{\mathbf{m}})),
\end{aligned}
\tag{4}
$$

where $W_l^{\overline{V}} \in \mathbb{R}^{C \times C_v}$ is a trainable projection matrix for the identification propagation. Particularly, the identification propagation shares the same attention maps, $Corr(I_l^t W_l^K, I_l^{\mathbf{m}} W_l^K)$, from the visual branch, since the identification of objects is mainly based on objects' visual features instead of their ID indices. Without the visual information, the tracking of objects is inapplicable.

## 4.2 Gated Propagation Module

Instead of using the LSTT block [63], which employs multi-head attention in propagation, we stack the hierarchical propagation based on the proposed Gated Propagation Module (GPM), which is designed based on more efficient single-head attention.

**LSTT Block** [63] includes four parts, *i.e.*, a long-term attention responsible for propagating information from the memorized frames (in $\mathbf{m}$), a short-term attention responsible for propagating information from a spatial neighborhood in the previous $(t-1)$ frame, a self-attention module for associating objects in the current $(t)$ frame, and a feed-forward module. The three kinds of attention modules are built on the multi-head [48] extension of Eq. 1 or Eq. 2. According to the experiments in Table 3b, reducing the head number from multiple heads (8 heads in default) to a single head will decrease the performance of AOT but can significantly improve the run-time speed, which means the multi-head attention is an efficiency bottleneck of LSTT. Concretely, the computational complexity of long-term attention is $\mathcal{O}(NTH^2W^2)$, which is proportional to the head number $N$ since each head contains a correlation function, $Corr(Q, K)$.

**Gated Propagation Function.** To avoid using multiple attention heads but not decrease the network performance, we redesign the attention-based VOS propagation defined in Eq. 1 and propose a gated propagation function as demonstrated in Fig. 3c. Let $U \in \mathbb{R}^{HW \times C}$ denotes a gating embedding, the function is

$$GP(U, Q, K, V) = \mathcal{F}_{dw}(\sigma(U) \odot Corr(Q, K)V)W^O, \tag{5}$$

where $\sigma$ is a non-linear gating function, $\odot$ denotes element-wise multiplication, $\mathcal{F}_{dw}(*)$ stands for a depth-wise 2D convolution layer [13], and $W^O \in \mathbb{R}^{C_v \times C}$ is the trainable weight of output projection. Firstly, we augment the attention-based propagation (Eq. 1) by using a conditional gate, $\sigma(U)$, which we empirically found to be effective in VOS. Notably, the presence of gating in weak attention mechanisms (*e.g.*, single-head attention) is also beneficial in some transformer-based methods [23, 29] for NLP. Moreover, we leverage a depth-wise convolution $\mathcal{F}_{dw}(*)$ to enhance the modeling of local spatial context in a lightweight manner.

**Gated Propagation Module** consists of three kinds of gated propagation, self-propagation, long-term propagation, and short-term propagation. Compared with LSTT, GPM removes the feed-forward module for further saving computation and parameters. All the propagation processes employ the gated propagation function defined in Eq. 5. In DeAOT, both the propagation branches (*i.e.*, visual branch and identification branch) are stacked by GPM as shown in Fig. 3b.

Based on the formulation of visual propagation (Eq. 3) and ID propagation (Eq. 4), the **Long-term Propagation** can be formulated as

$$GP_{lt}^{vis}(I_l^t, I_l^t, I_l^{\mathbf{m}}, I_l^{\mathbf{m}}) = GP(I_l^t W_l^U, I_l^t W_l^K, I_l^{\mathbf{m}} W_l^K, I_l^{\mathbf{m}} W_l^V), \tag{6}$$

$$GP_{lt}^{id}(M_l^t, I_l^t, I_l^{\mathbf{m}}, M_l^{\mathbf{m}}, Y^{\mathbf{m}}) = GP(M_l^t W_l^{\overline{U}}, I_l^t W_l^K, I_l^{\mathbf{m}} W_l^K, M_l^{\mathbf{m}} W_l^{\overline{V}} + ID(Y^{\mathbf{m}})) \tag{7}$$

for the visual branch and ID branch, respectively. The ID propagation reuses the attention maps of the visual propagation as discussed in Eq. 4. Based on the long-term propagation, we can formulate the **Short-term Propagation** at spatial location $p$ to be

$$GP_{st}^{vis}(I_l^t, I_l^t, I_l^{t-1}, I_l^{t-1}|p) = GP_{lt}^{vis}(I_{l,p}^t, I_{l,p}^t, I_{l,\mathcal{N}(p)}^{t-1}, I_{l,\mathcal{N}(p)}^{t-1}), \tag{8}$$

$$GP_{st}^{id}(M_l^t, I_l^t, I_l^{t-1}, M_l^{t-1}, Y^{t-1}|p) = GP_{lt}^{id}(M_{l,p}^t, I_{l,p}^t, I_{l,\mathcal{N}(p)}^{t-1}, M_{l,\mathcal{N}(p)}^{t-1}|Y_{\mathcal{N}(p)}^{t-1}), \tag{9}$$

where $I_{l,p}^t, M_{l,p}^t \in \mathbb{R}^{1 \times C}$ are the feature of $I_l^t, M_l^t$ at location $p$ respectively, and $\mathcal{N}(p)$ stands for a $\lambda \times \lambda$ spatial neighbourhood centered at location $p$. The short-term propagation for each location $p$ is restricted in its spatial neighbourhood ($I_{l,\mathcal{N}(p)}^{t-1}$ or $M_{l,\mathcal{N}(p)}^{t-1}$) of the previous $(t-1)$ frame. Since the object motions across several contiguous video frames are always smooth, non-local propagation processes becomes inefficient and not necessary in short-term information propagation [62].

Finally, the **Self-Propagation** can also be formulated similar to the long-term propagation, *i.e.*,

$$GP_{self}^{vis}(I_l^t|M_l^t) = GP(I_l^t W_l^U, (I_l^t \oplus M_l^t)W_l^K, (I_l^t \oplus M_l^t)W_l^K, I_l^t W_l^V), \tag{10}$$

$$GP_{self}^{id}(M_l^t|I_l^t) = GP(M_l^t W_l^{\overline{U}}, (I_l^t \oplus M_l^t)W_l^K, (I_l^t \oplus M_l^t)W_l^K, M_l^t W_l^{\overline{V}}), \tag{11}$$

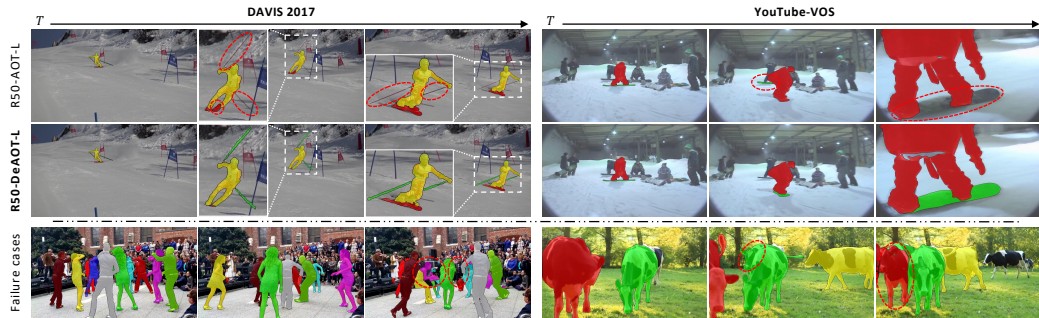

Figure 4: Qualitative results. (top) DeAOT performs better than AOT [63] on tiny or scale-changing objects. (bottom) DeAOT fails to track highly similar objects when serious occlusion happens.

where $\oplus$ is a concatenation process on the channel dimension. In the self-propagations, both the visual embedding $I_l^t$ and ID embedding $M_l^t$ are used in the calculation of attention maps (*i.e.*, $Corr(Q, K)$). Here, the object-specific $M_l^t$ performs like a positional embedding [48] additional to the visual embedding $I_l^t$. We found that such a process can help associate the objects in the current frame more effectively. Apart from this, the current frame segmentation $Y^t$ is unavailable before being decoded and is not used in the ID self-propagation $GP_{self}^{id}$. For simplicity, we reuse the parameter symbols in Eq. 6 and 7, but the trainable parameters are not shared with long-term propagation.

## 5 Implementation Details

**Network Details:** Consistent with AOT [63], three kinds of encoders are used in our experiments, *i.e.*, MobileNet-V2 [42] (in default), ResNet-50 (R50) [21], and Swin-B [30]. The decoder is the same FPN [27] network. Besides, the spatial neighborhood size $\lambda$ is set to 15, and the maximum object number within the ID embedding is 10. In our GPM module, the channel dimension $C$ of visual and ID embeddings is 256, the matching features' dimension $C_k$ is 128, and the propagation features' dimension $C_v$ is 512. Moreover, the kernel size of $\mathcal{F}_{dw}$ is 5, and the gating function $\sigma(*)$ is SiLU/Swish [18, 41].

To make fair comparisons with AOT's variants [63], we build corresponding DeAOT variants with different GPM number $L$ or long-term memory size $\mathbf{m}$. The hyper-parameters of these variants are: **DeAOT-T**: $L = 1$, $\mathbf{m} = \{1\}$; **DeAOT-S**: $L = 2$, $\mathbf{m} = \{1\}$; **DeAOT-B**: $L = 3$, $\mathbf{m} = \{1\}$; **DeAOT-L**: $L = 3$, $\mathbf{m} = \{1, 1+\delta, 1+2\delta, ...\}$. DeAOT-T/S/B considers only the reference frame as the long-term memory, leading to consistent run-time speeds. DeAOT-L updates the long-term memory per $\delta$ (set to 2/5 for training/testing) frames as AOT-L [63].

**Training Details:** Following [34, 43, 44, 55, 63], we first pre-train DeAOT on synthetic video sequence generated from static image datasets [12, 19, 20, 28, 45] by randomly applying multiple image augmentations [55]. Then, we do main training on the VOS benchmarks [39, 57] by randomly applying video augmentations [62, 63]. Besides, we keep our optimization strategies and related hyper-parameters the same as AOT. More details are supplied in Supplementary.

## 6 Experimental Results

We conduct experiments on three popular VOS benchmarks (YouTube-VOS [57], DAVIS 2017 [39], and DAVIS 2016 [38]) and one challenging Visual Object Tracking (VOT) benchmark (VOT 2020 [24]), which gives segmentation annotations and can be used to evaluate VOS algorithms.

To validate DeAOT's generalization ability, all the benchmarks share the same model parameters. When evaluating YouTube-VOS, we use the default 6fps videos, which are restricted to be smaller than $1.3 \times 480p$ resolution. On DAVIS, the default 480p 24fps videos are used. For evaluating VOT 2020, more details can be found in the supplementary material.

The evaluation metrics for VOS benchmarks include the $\mathcal{J}$ score (calculated as the average IoU score between the prediction and the ground truth mask), the $\mathcal{F}$ score (calculated as an average boundary

Table 1: The quantitative evaluation on multi-object benchmarks, YouTube-VOS [57] and DAVIS 2017 [39]. $\mathcal{J}_S/\mathcal{F}_S/\mathcal{J}_U/\mathcal{F}_U$: $\mathcal{J}/\mathcal{F}$ on seen/unseen classes. ‡: timing extrapolated from single-object speed assuming linear scaling in the number of objects. ⋆: recorded on our device.

| Method | YouTube-VOS 2018 Val | | | | | YouTube-VOS 2019 Val | | | | | | DAVIS-17 Val | | | DAVIS-17 Test | | | |
|---|---|---|---|---|---|---|---|---|---|---|---|---|---|---|---|---|---|---|
| | Avg | $\mathcal{J}_S$ | $\mathcal{F}_S$ | $\mathcal{J}_U$ | $\mathcal{F}_U$ | Avg | $\mathcal{J}_S$ | $\mathcal{F}_S$ | $\mathcal{J}_U$ | $\mathcal{F}_U$ | fps | Avg | $\mathcal{J}$ | $\mathcal{F}$ | Avg | $\mathcal{J}$ | $\mathcal{F}$ | fps |
| KMN[ECCV20] [43] | 81.4 | 81.4 | 85.6 | 75.3 | 83.3 | - | - | - | - | - | - | 82.8 | 80.0 | 85.6 | 77.2 | 74.1 | 80.3 | - |
| CFBI[ECCV20] [62] | 81.4 | 81.1 | 85.8 | 75.3 | 83.4 | 81.0 | 80.6 | 85.1 | 75.2 | 83.0 | 3.4 | 81.9 | 79.3 | 84.5 | 76.6 | 73.0 | 80.1 | 2.9 |
| SST[CVPR21] [17] | 81.7 | 81.2 | - | 76.0 | - | 81.8 | 80.9 | - | 76.6 | - | - | 82.5 | 79.9 | 85.1 | - | - | - | - |
| HMMN[ICCV21] [44] | 82.6 | 82.1 | 87.0 | 76.8 | 84.6 | 82.5 | 81.7 | 86.1 | 77.3 | 85.0 | - | 84.7 | 81.9 | 87.5 | 78.6 | 74.7 | 82.5 | 3.4‡ |
| CFBI+[TPAMI21] [64] | 82.8 | 81.8 | 86.6 | 77.1 | 85.6 | 82.6 | 81.7 | 86.2 | 77.1 | 85.2 | 4.0 | 82.9 | 80.1 | 85.7 | 78.0 | 74.4 | 81.6 | 3.4 |
| STCN[NeurIPS21] [11] | 83.0 | 81.9 | 86.5 | 77.9 | 85.7 | 82.7 | 81.1 | 85.4 | 78.2 | 85.9 | 8.4⋆ | 85.4 | 82.2 | 88.6 | 76.1 | 72.7 | 79.6 | 19.5⋆ |
| RPCM[AAAI22] [58] | 84.0 | 83.1 | 87.7 | 78.5 | 86.7 | 83.9 | 82.6 | 86.9 | 79.1 | 87.1 | - | 83.7 | 81.3 | 86.0 | 79.2 | 75.8 | 82.6 | - |
| AOT-T [63] | 80.2 | 80.1 | 84.5 | 74.0 | 82.2 | 79.7 | 79.6 | 83.8 | 73.7 | 81.8 | 41.0 | 79.9 | 77.4 | 82.3 | 72.0 | 68.3 | 75.7 | 51.4 |
| DeAOT-T | **82.0** | **81.6** | **86.3** | **75.8** | **84.2** | **82.0** | **81.2** | **85.6** | **76.4** | **84.7** | **53.4** | **80.5** | **77.7** | **83.3** | **73.7** | **70.0** | **77.3** | **63.5** |
| AOT-S [63] | 82.6 | 82.0 | 86.7 | 76.6 | 85.0 | 82.2 | 81.3 | 85.9 | 76.6 | 84.9 | 27.1 | **81.3** | **78.7** | **83.9** | 73.9 | 70.3 | 77.5 | 40.0 |
| DeAOT-S | **84.0** | **83.3** | **88.3** | **77.9** | **86.6** | **83.8** | **82.8** | **87.5** | **78.1** | **86.8** | **38.7** | 80.8 | 77.8 | 83.8 | **75.4** | **71.9** | **79.0** | **49.2** |
| AOT-B [63] | 83.5 | 82.6 | 87.5 | 77.7 | 86.0 | 83.3 | 82.4 | 87.1 | 77.8 | 86.0 | 20.5 | **82.5** | **79.7** | **85.2** | 75.5 | 71.6 | 79.3 | 29.6 |
| DeAOT-B | **84.6** | **83.9** | **88.9** | **78.5** | **87.0** | **84.6** | **83.5** | **88.3** | **79.1** | **87.5** | **30.4** | 82.2 | 79.2 | 85.1 | **76.2** | **72.5** | **79.9** | **40.9** |
| AOT-L [63] | 83.8 | 82.9 | 87.9 | 77.7 | 86.5 | 83.7 | 82.8 | 87.5 | 78.0 | 86.7 | 16.0 | 83.8 | **81.1** | 86.4 | **78.3** | **74.3** | **82.3** | 18.7 |
| DeAOT-L | **84.8** | **84.2** | **89.4** | **78.6** | **87.0** | **84.7** | **83.8** | **88.8** | **79.0** | **87.2** | **24.7** | **84.1** | 81.0 | **87.1** | 77.9 | 74.1 | 81.7 | **28.5** |
| R50-AOT-L [63] | 84.1 | 83.7 | 88.5 | 78.1 | 86.1 | 84.1 | 83.5 | 88.1 | 78.4 | 86.3 | 14.9 | 84.9 | **82.3** | 87.5 | 79.6 | 75.9 | 83.3 | 18.0 |
| R50-DeAOT-L | **86.0** | **84.9** | **89.9** | **80.4** | **88.7** | **85.9** | **84.6** | **89.4** | **80.8** | **88.9** | **22.4** | **85.2** | 82.2 | **88.2** | **80.7** | **76.9** | **84.5** | **27.0** |
| SwinB-AOT-L [63] | 84.5 | 84.3 | 89.3 | 77.9 | 86.4 | 84.5 | 84.0 | 88.8 | 78.4 | 86.7 | 9.3 | 85.4 | 82.4 | 88.4 | 81.2 | 77.3 | 85.1 | 12.1 |
| SwinB-DeAOT-L | **86.2** | **85.6** | **90.6** | **80.0** | **88.4** | **86.1** | **85.3** | **90.2** | **80.4** | **88.6** | **11.9** | **86.2** | **83.1** | **89.2** | **82.8** | **78.9** | **86.7** | **15.4** |

similarity measure between the boundary of the prediction and the ground truth), and their mean value (denoted as $\mathcal{J}\&\mathcal{F}$). As to VOT 2020, we use the official EAO criteria [24]. We evaluate all the results on official evaluation servers or with official tools.

## 6.1 Compare with the State-of-the-art Methods

**YouTube-VOS** [57] is a large-scale multi-object VOS benchmark, which contains 3471 videos in the training split with 65 categories and 474/507 videos in the Validation 2018/2019 split with additional 26 unseen categories. Table 1 shows that DeAOT variants remarkably outperforms AOT counterparts in both accuracy and run-time speed on YouTube-VOS 2018/2019. For example, our R50-DeAOT-L achieves **86.0%/85.9%** ($\mathcal{J}\&\mathcal{F}$) at **22.4fps**, which is superior compared to R50-AOT-L [63] (84.1%/84.1% at 14.9fps). Particularly, our SwinB-DeAOT-L achieves new state-of-the-art performance (**86.2%/86.1%**), surpassing previous methods by more than 1.7%/1.6%. In addition, our smallest variant, DeAOT-T, precedes SST [17] (**82.0%/82.0%** vs 81.7%/81.8%) and runs about **15×** faster than CFBI [62] (**53.4fps** vs 3.4fps).

**DAVIS 2017** [39] is a multi-object extension of DAVIS 2016. The training/validation split consists of 60/30 videos with 138/59 objects, and the test split contains 30 more challenging videos with 89 objects. As shown in Table 1, DeAOT variants can generalize to DAVIS 2017 well. R50-DeAOT-L achieves **85.2%/80.7%** on the validation/test split at a real-time speed (**27fps**), surpassing R50-AOT-L in accuracy and efficiency. Also, SwinB-DeAOT-L achieves the top-ranked performance on DAVIS 2017 (**86.2%/82.8%**).

**DAVIS 2016** [38] is a single-object benchmark containing 20 videos in the validation split, and we show related experiments in Table 2. Although AOT-like methods focus on multi-object scenarios, our DeAOT-L is faster and more robust than STCN [11], whose architecture was designed for single-object VOS. Besides, SwinB-DeAOT-L achieves **92.9%** and outperforms all the VOS methods as well.

Table 2: The quantitative evaluation on the single-object benchmarks, DAVIS 2016 [38] and VOT 2020 [24]. EAO$^{RT}$: real-time EAO metric [24].

| Method | DAVIS 2016 | | | | VOT 2020 | |
|---|---|---|---|---|---|---|
| | Avg | $\mathcal{J}$ | $\mathcal{F}$ | fps | EAO | EAO$^{RT}$ |
| CFBI+ [64] | 89.9 | 88.7 | 91.1 | 5.9 | - | - |
| RPCM [58] | 90.6 | 87.1 | 94.0 | 5.8 | - | - |
| HMMN [44] | 90.8 | 89.6 | 92.0 | 10.0 | - | - |
| STCN [11] | 91.6 | 90.8 | 92.5 | 27.2⋆ | - | - |
| AlphaRef [59] | - | - | - | - | 0.482 | 0.486 |
| RPT [33] | - | - | - | - | 0.530 | 0.290 |
| MixFormer-L [14] | - | - | - | - | 0.555 | - |
| AOT-T [63] | 86.8 | 86.1 | 87.4 | 51.4 | 0.435 | 0.433 |
| DeAOT-T | **88.9** | **87.8** | **89.9** | **63.5** | **0.472** | **0.463** |
| AOT-S [63] | **89.4** | **88.6** | 90.2 | 40.0 | 0.512 | 0.499 |
| DeAOT-S | 89.3 | 87.6 | **90.9** | **49.2** | **0.593** | **0.559** |
| AOT-B [63] | 89.9 | 88.7 | 91.1 | 29.6 | 0.541 | 0.533 |
| DeAOT-B | **91.0** | **89.4** | **92.5** | **40.9** | **0.571** | **0.542** |
| AOT-L [63] | 90.4 | 89.6 | 91.1 | 18.7 | 0.574 | 0.560 |
| DeAOT-L | **92.0** | **90.3** | **93.7** | **28.5** | **0.591** | **0.554** |
| R50-AOT-L [63] | 91.1 | 90.1 | 92.1 | 18.0 | 0.569 | 0.540 |
| R50-DeAOT-L | **92.3** | **90.5** | **94.0** | **27.0** | **0.613** | **0.571** |
| SwinB-AOT-L [63] | 92.0 | 90.7 | 93.3 | 12.1 | 0.586 | 0.523 |
| SwinB-DeAOT-L | **92.9** | **91.1** | **94.7** | **15.4** | **0.622** | **0.559** |

Table 3: Ablation study. The experiments are conducted on YouTube-VOS 2018 [57] and based on DeAOT-S without pre-training on static images. De: decoupling features. $C$: the channel dimension. Prop: propagation type. LT/ST: long-term/short-term. $ks$: kernel size.

(a) Propagation module

| Module | $C$ | $\mathcal{J}\&\mathcal{F}$ | $\mathcal{J}_S$ | $\mathcal{J}_U$ |
|---|---|---|---|---|
| **GPM** | 256 | **82.5** | **82.3** | **76.1** |
| *w/o* De | 256 | 81.5 | 81.4 | 75.0 |
| *w/o* De | 512 | 82.0 | 82.1 | 75.4 |
| LSTT | 256 | 80.3 | 80.6 | 73.7 |

(b) Head number ($N_h$)

| Model | $N_h$ | $\mathcal{J}\&\mathcal{F}$ | $\mathcal{J}_S$ | $\mathcal{J}_U$ | fps |
|---|---|---|---|---|---|
| **DeAOT** | 1 | **82.5** | **82.3** | **76.1** | 38.7 |
| DeAOT | 8 | **82.5** | **82.3** | 75.8 | 24.7 |
| AOT | 1 | 79.6 | 80.1 | 72.6 | **44.6** |
| AOT | 8 | 80.3 | 80.6 | 73.7 | 27.1 |

(c) Attention map

| Prop | Vis | ID | $\mathcal{J}\&\mathcal{F}$ | $\mathcal{J}_S$ | $\mathcal{J}_U$ |
|---|---|---|---|---|---|
| **LT/ST** | ✓ | | **82.5** | **82.3** | **76.1** |
| LT/ST | ✓ | ✓ | 82.1 | 82.2 | 75.7 |
| **Self** | ✓ | ✓ | **82.5** | **82.3** | **76.1** |
| Self | ✓ | | 82.2 | 82.1 | 75.7 |

(d) $ks$ of $\mathcal{F}_{dw}$

| $ks$ | $\mathcal{J}\&\mathcal{F}$ | $\mathcal{J}_S$ | $\mathcal{J}_U$ |
|---|---|---|---|
| 5 | **82.5** | **82.3** | **76.1** |
| 0 | 81.1 | 81.5 | 74.2 |
| 3 | 82.2 | 82.2 | 76.1 |
| 9 | 82.4 | 82.2 | 75.8 |

**VOT 2020** [24] consists of 60 single-object videos with challenging scenarios including fast motion, occlusion, etc. The average frame number of VOT 2020 is 327, which is much longer than the maximum video length of the above VOS benchmarks. DeAOT shows superior performance on VOT 2020 in Table 2. The DeAOT variants larger than DeAOT-T outperform MixFormer-L [14] (the state-of-the-art tracker), RPT [33] (VOT 2020 short-term challenge winner), and AlphaRef [59] (VOT 2020 real-time challenge winner) in both EAO and real-time EAO scores. Specifically, SwinB-DeAOT-L achieves **0.622** EAO, outstandingly exceeding MixFormer-L by **0.067**, and R50-DeAOT-L achieves **0.571** EAO under a **real-time** requirement, impressively overtaking AlphaRef by **0.085**.

**Qualitative results:** Fig. 4 give qualitative comparisons to AOT. By introducing the dual-branch propagation, R50-DeAOT-L performs better than R50-AOT-L on tiny or scale-changing objects (*ski poles* or *ski board*). Nevertheless, R50-DeAOT-L still may fails to track multiple highly similar objects (*dancer* and *cow*) when serious occlusion happens.

## 6.2 Ablation Study

This section analyzes the necessity of dual-branch propagation and GPM of DeAOT in Table 3.

**Propagation module:** Table 3a shows that the performance of DeAOT drops from 82.5% to 81.5% by coupling the propagation of visual and ID embeddings (*w/o* De) like AOT. Furthermore, doubling the channel dimensions only partially relieves the performance loss. Moreover, the performance will be seriously degraded to 80.3% by replacing our GPM with the LSTT module of AOT. In conclusion, the dual-branch propagation approach and the GPM module are crucial in improving VOS performance.

**Head number:** According to the results in Table 3b, the head number ($N_h$) of attention-based modules is negatively correlated with the efficiency of AOT/De-AOT. The single-head AOT (44.6fps) runs much faster than the default AOT ($N_h$=8, 27.1fps) but loses 0.7% accuracy. By contrast, DeAOT is robust to the head number by using our proposed GPM module.

**Attention map:** Our DeAOT shares the attention maps between two propagation branches. Table 3c shows the study of different kinds of attention maps. Concretely, visual embeddings are essential in building attention maps in the long-term/short-term propagation, whose attention maps are used to match objects. Introducing ID embeddings does not help learn better visual embeddings and will decrease the performance (82.5% *vs* 82.1%). In the self-propagation, however, utilizing the ID embedding as a positional embedding will facilitate the association of objects (82.2% *vs* 82.5%) in the current frame.

**Kernel size of $\mathcal{F}_{dw}$:** Large receptive fields have been proved to be critical in segmentation-related tasks [9]. The depth-wise convolution, $\mathcal{F}_{dw}$, is an important part of GPM for enlarging the receptive fields. Without $\mathcal{F}_{dw}$, the performance of DeAOT drops from 82.5% to 81.1%, as shown in Table 3d. We empirically found the best kernel size of $\mathcal{F}_{dw}$ is 5 among $\{3, 5, 9\}$.

## 7 Conclusion

This paper proposes a highly effective and efficient framework, Decoupling Features in Hierarchical Propagation (DeAOT), for video object segmentation. Based on the rethinking of AOT-like hierarchical propagation, we propose to decouple the propagation of visual and ID embeddings into two network branches and thus avoid the loss of visual information in deep propagation layers. Besides,

we propose the Gated Propagation Module (GPM), an efficient module for constructing hierarchical VOS propagation. Applying GPM to the dual-branch propagation, our DeAOT variant networks achieve new state-of-the-art performance on four VOS/VOT benchmarks with superior run-time speed compared to previous solutions.

**Acknowledgements.** This work is partly supported by the Fundamental Research Funds for the Central Universities (No. 226-2022-00051).

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
