# Supplementary Materials of
# Decoupling Features in Hierarchical Propagation for Video Object Segmentation

**Zongxin Yang**[1,2]**, Yi Yang**[1]

[1] CCAI, College of Computer Science and Technology, Zhejiang University  [2] Baidu Research
{yangzongxin, yangyics}@zju.edu.cn

## A  Appendix

### A.1  More Training Details

Following [3, 13, 16, 19, 24, 28], we first pre-train DeAOT on synthetic video sequence generated from static image datasets [4, 6, 7, 11, 21] by randomly applying multiple image augmentations [24]. Then, we do main training on the VOS benchmarks [18, 25] by randomly applying video augmentations [3, 27, 28].

The optimization strategies and related hyper-parameters are also the same as AOT. In detail, we adopt the AdamW [12] optimizer and the sequential training strategy [27] with a sequence length of 5. The loss function is a 0.5:0.5 combination of BCE loss [22] and soft Jaccard loss [15]. For pre-training, we use an initial learning rate of $4 \times 10^{-4}$ and a weight decay of 0.03. For main training, the initial learning rate is set to $2 \times 10^{-4}$, and the weight decay is 0.07. Each training stage takes 100,000 steps, and the batch size is 16. We used 4/1 Tesla V100 GPU for training/testing.

### A.2  Inference Details on VOT 2020

Most of the inference details on VOT 2020 [9] are the same as the inference setting of AOT [28] on YouTube-VOS [25] and DAVIS [17, 18]. The differences are listed below: (1) The size of input videos is resized to be smaller than $1.3\times480p$ and larger than 480p, since some VOT-2020 videos are smaller than 480p, which is too small to extract object features effectively. (2) We update the long-term memory DeAOT/AOT per 10 frames instead of 5 frames, and we will drop the oldest frame (except for the reference frame) from the long-term memory if the memory size is larger than 10 frames. In other words, the maximum temporal range of long-term memory is $10 \times 10 = 100$ frames. Such a process is necessary to keep enough long-term information and avoid facing out of memory when inferring long videos. The longest video in VOT 2020 contains 1,500 frames. (3) When processing tiny objects smaller than 1/900 of the video size. We conduct video object segmentation only in a small cropped window, which is dynamically centered at the object's position in the last frame where the object is not occluded. The size of cropped window is only 1/12 of the input resolution, and the cropped image will be resized to $465\times465$. By doing this, we can further save computations in the inference stage and will not lose performance since tiny objects always move slowly.

### A.3  Compare with More VOS Methods

We compare our DeAOT with more VOS methods in Table 2 and 1. As shown, DeAOT variant networks outperform all the competitors in both performance and efficiency.

36th Conference on Neural Information Processing Systems (NeurIPS 2022).

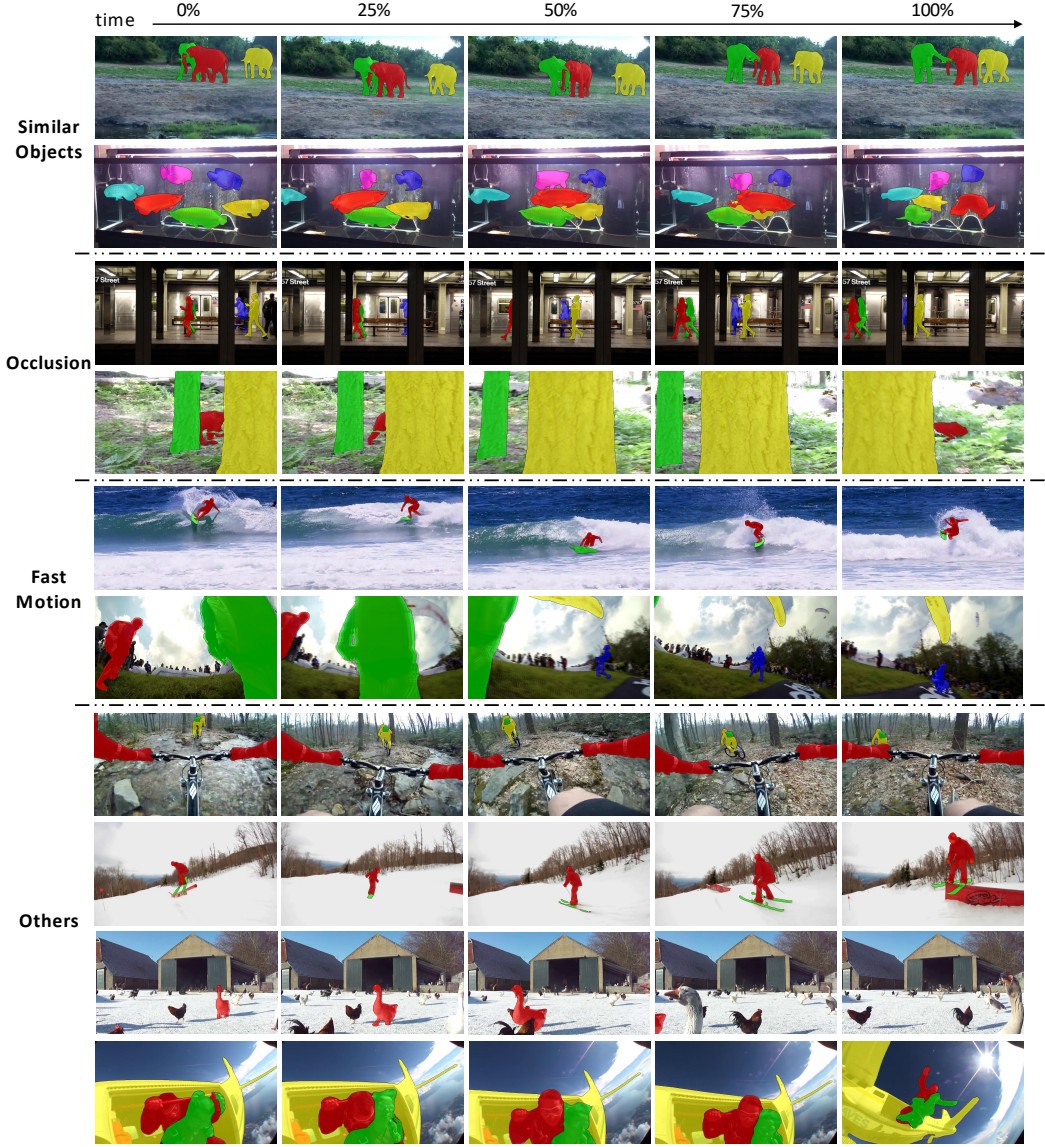

Figure 1: Qualitative results on YouTube-VOS [25] and DAVIS[18]. R50-DeAOT-L tracks and segments objects well under many challenging scenarios, including similar objects, occlusion, fast motion, etc.

We also supply more qualitative results under challenging scenarios on YouTube-VOS [25] and DAVIS 2017 [18] in Fig. 1, respectively. As demonstrated, DeAOT is robust to many challenging VOS cases, including similar objects, occlusion, fast motion, motion blur, etc.

## A.4 Border Impact and Limitations

The proposed DeAOT framework significantly improves VOS's performance, robustness, and robustness. The DeAOT variants with real-time speeds may benefit the applications of VOS in real-time video systems, such as video conference, self-driving car, augmented reality, etc. Also, DeAOT may be used in video surveillance systems for short-term and precise object tracking, although this is not our target requirement to promote the development of the VOS community.

As to limitations, the scenarios with multiple similar objects and severe occlusions are still very challenging for DeAOT and other VOS solutions. Besides, there are rare studies about VOS of long-term videos containing thousands of frames since the VOS community has no high-quality

Table 1: Additional quantitative comparison on DAVIS 2016 [17].

| Methods | $\mathcal{J}\&\mathcal{F}$ | $\mathcal{J}$ | $\mathcal{F}$ | FPS |
|---|---|---|---|---|
| FEEL [22] | 81.7 | 81.1 | 82.2 | 2.2 |
| AG [8] | 82.1 | 82.2 | 82.0 | 14.3 |
| SAT [2] | 83.1 | 82.6 | 83.6 | 39 |
| STM [16] | 89.3 | 88.7 | 89.9 | 6.3 |
| CFBI [27] | 89.4 | 88.3 | 90.5 | 6.3 |
| CFBI+ [29] | 89.9 | 88.7 | 91.1 | 5.9 |
| KMN [19] | 90.5 | 89.5 | 91.5 | 8.3 |
| RPCM [26] | 90.6 | 87.1 | 94.0 | 5.8 |
| HMMN [20] | 90.8 | 89.6 | 92.0 | 10.0 |
| STCN [3] | 91.6 | 90.8 | 92.5 | 27.2* |
| AOT-T [28] | 86.8 | 86.1 | 87.4 | 51.4 |
| DeAOT-T | **88.9** | **87.8** | **89.9** | **63.5** |
| AOT-S [28] | **89.4** | **88.6** | 90.2 | 40.0 |
| DeAOT-S | 89.3 | 87.6 | **90.9** | **49.2** |
| AOT-B [28] | 89.9 | 88.7 | 91.1 | 29.6 |
| DeAOT-B | **91.0** | **89.4** | **92.5** | **40.9** |
| AOT-L [28] | 90.4 | 89.6 | 91.1 | 18.7 |
| DeAOT-L | **92.0** | **90.3** | **93.7** | **28.5** |
| R50-AOT-L [28] | 91.1 | 90.1 | 92.1 | 18.0 |
| R50-DeAOT-L | **92.3** | **90.5** | **94.0** | **27.0** |
| SwinB-AOT-L [28] | 92.0 | 90.7 | 93.3 | 12.1 |
| SwinB-DeAOT-L | **92.9** | **91.1** | **94.7** | **15.4** |

dataset for long-term video segmentation. However, applications (*e.g.*, augmented reality) in the real world often require the algorithms to have the ability to process long videos smoothly.

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

Table 2: Additional quantitative comparison on multi-object benchmarks, YouTube-VOS [25] and DAVIS 2017 [18]. *: using 600p instead of 480p videos in inference. ‡: timing extrapolated from single-object speed assuming linear scaling in the number of objects. ⋆: recorded on our device.

(a) YouTube-VOS

| Methods | Seen | | | Unseen | | FPS |
|---|---|---|---|---|---|---|
| | $\mathcal{J}\&\mathcal{F}$ | $\mathcal{J}$ | $\mathcal{F}$ | $\mathcal{J}$ | $\mathcal{F}$ | |
| *Validation 2018 Split* | | | | | | |
| SAT[CVPR20] [2] | 63.6 | 67.1 | 70.2 | 55.3 | 61.7 | - |
| AG[CVPR19] [8] | 66.1 | 67.8 | - | 60.8 | - | - |
| PReM[ACCV18] [14] | 66.9 | 71.4 | 75.9 | 56.5 | 63.7 | 0.17 |
| BoLT[arXiv19] [23] | 71.1 | 71.6 | - | 64.3 | - | 0.74 |
| GC[ECCV20] [10] | 73.2 | 72.6 | 75.6 | 68.9 | 75.7 | - |
| STM[ICCV19] [16] | 79.4 | 79.7 | 84.2 | 72.8 | 80.9 | - |
| EGMN[ECCV20] [13] | 80.2 | 80.7 | 85.1 | 74.0 | 80.9 | - |
| KMN[ECCV20] [19] | 81.4 | 81.4 | 85.6 | 75.3 | 83.3 | - |
| CFBI[ECCV20] [27] | 81.4 | 81.1 | 85.8 | 75.3 | 83.4 | 3.4 |
| LWL[ECCV20] [1] | 81.5 | 80.4 | 84.9 | 76.4 | 84.4 | - |
| SST[CVPR21] [5] | 81.7 | 81.2 | - | 76.0 | - | - |
| CFBI+[TPAMI21] [29] | 82.8 | 81.8 | 86.6 | 77.1 | 85.6 | 4.0 |
| HMMN[ICCV21] [20] | 82.6 | 82.1 | 87.0 | 76.8 | 84.6 | - |
| STCN[NeurIPS21] [3] | 83.0 | 81.9 | 86.5 | 77.9 | 85.7 | 8.4⋆ |
| RPCM[AAAI22] [26] | 84.0 | 83.1 | 87.7 | 78.5 | 86.7 | - |
| AOT-T [28] | 80.2 | 80.1 | 84.5 | 74.0 | 82.2 | 41.0 |
| DeAOT-T | **82.0** | **81.6** | **86.3** | **75.8** | **84.2** | **53.4** |
| AOT-S [28] | 82.6 | 82.0 | 86.7 | 76.6 | 85.0 | 27.1 |
| DeAOT-S | **84.0** | **83.3** | **88.3** | **77.9** | **86.6** | **38.7** |
| AOT-B [28] | 83.5 | 82.6 | 87.5 | 77.7 | 86.0 | 20.5 |
| DeAOT-B | **84.6** | **83.9** | **88.9** | **78.5** | **87.0** | **30.4** |
| AOT-L [28] | 83.8 | 82.9 | 87.9 | 77.7 | 86.5 | 16.0 |
| DeAOT-L | **84.8** | **84.2** | **89.4** | **78.6** | **87.0** | **24.7** |
| R50-AOT-L [28] | 84.1 | 83.7 | 88.5 | 78.1 | 86.1 | 14.9 |
| R50-DeAOT-L | **86.0** | **84.9** | **89.9** | **80.4** | **88.7** | **22.4** |
| SwB-AOT-L [28] | 84.5 | 84.3 | 89.3 | 77.9 | 86.4 | 9.3 |
| SwB-DeAOT-L | **86.2** | **85.6** | **90.6** | **80.0** | **88.4** | **11.9** |
| *Validation 2019 Split* | | | | | | |
| CFBI[ECCV20] [27] | 81.0 | 80.6 | 85.1 | 75.2 | 83.0 | 3.4 |
| SST[CVPR21] [5] | 81.8 | 80.9 | - | 76.6 | - | - |
| HMMN[ICCV21] [20] | 82.5 | 81.7 | 86.1 | 77.3 | 85.0 | - |
| CFBI+[TPAMI21] [29] | 82.6 | 81.7 | 86.2 | 77.1 | 85.2 | 4.0 |
| STCN[NeurIPS21] [3] | 82.7 | 81.1 | 85.4 | 78.2 | 85.9 | 8.4⋆ |
| RPCM[AAAI22] [26] | 83.9 | 82.6 | 86.9 | 79.1 | 87.1 | - |
| AOT-T [28] | 79.7 | 79.6 | 83.8 | 73.7 | 81.8 | 41.0 |
| DeAOT-T | **82.0** | **81.2** | **85.6** | **76.4** | **84.7** | **53.4** |
| AOT-S [28] | 82.2 | 81.3 | 85.9 | 76.6 | 84.9 | 27.1 |
| DeAOT-S | **83.8** | **82.8** | **87.5** | **78.1** | **86.8** | **38.7** |
| AOT-B [28] | 83.3 | 82.4 | 87.1 | 77.8 | 86.0 | 20.5 |
| DeAOT-B | **84.6** | **83.5** | **88.3** | **79.1** | **87.5** | **30.4** |
| AOT-L [28] | 83.7 | 82.8 | 87.5 | 78.0 | **86.7** | 16.0 |
| DeAOT-L | **84.7** | **83.8** | **88.8** | **79.0** | 87.2 | **24.7** |
| R50-AOT-L [28] | 84.1 | 83.5 | 88.1 | 78.4 | 86.3 | 14.9 |
| R50-DeAOT-L | **85.9** | **84.6** | **89.4** | **80.8** | **88.9** | **22.4** |
| SwB-AOT-L [28] | 84.5 | 84.0 | 88.8 | 78.4 | 86.7 | 9.3 |
| SwB-DeAOT-L | **86.1** | **85.3** | **90.2** | **80.4** | **88.6** | **11.9** |

(b) DAVIS 2017

| Methods | $\mathcal{J}\&\mathcal{F}$ | $\mathcal{J}$ | $\mathcal{F}$ | FPS |
|---|---|---|---|---|
| *Validation 2017 Split* | | | | |
| AG [8] | 70.0 | 67.2 | 72.7 | 7.1‡ |
| FEEL [22] | 71.5 | 69.1 | 74.0 | 2.0 |
| SAT [2] | 72.3 | 68.6 | 76.0 | 19.5‡ |
| LWL [1] | 81.6 | 79.1 | 84.1 | 2.5‡ |
| STM [16] | 81.8 | 79.2 | 84.3 | 3.1‡ |
| CFBI [27] | 81.9 | 79.3 | 84.5 | 5.9 |
| SST [5] | 82.5 | 79.9 | 85.1 | - |
| EGMN [13] | 82.8 | 80.2 | 85.2 | 2.5‡ |
| KMN [19] | 76.0 | 74.2 | 77.8 | 4.2‡ |
| KMN [19] | 82.8 | 80.0 | 85.6 | 4.2‡ |
| CFBI+ [29] | 82.9 | 80.1 | 85.7 | 5.6 |
| SST [5] | 82.5 | 79.9 | 85.1 | - |
| KMN [19] | 82.8 | 80.0 | 85.6 | 4.2‡ |
| CFBI+ [29] | 82.9 | 80.1 | 85.7 | 5.6 |
| RPCM [26] | 83.7 | 81.3 | 86.0 | - |
| HMMN [20] | 84.7 | 81.9 | 87.5 | 5.0‡ |
| STCN [3] | 85.4 | 82.2 | 88.6 | 24.7⋆ |
| AOT-T [28] | 79.9 | 77.4 | 82.3 | 51.4 |
| DeAOT-T | **80.5** | **77.7** | **83.3** | **63.5** |
| AOT-S [28] | **81.3** | **78.7** | **83.9** | 40.0 |
| DeAOT-S | 80.8 | 77.8 | 83.8 | **49.2** |
| AOT-B [28] | **82.5** | **79.7** | **85.2** | 29.6 |
| DeAOT-B | 82.2 | 79.2 | 85.1 | **40.9** |
| AOT-L [28] | 83.8 | **81.1** | 86.4 | 18.7 |
| DeAOT-L | **84.1** | 81.0 | **87.1** | **28.5** |
| R50-AOT-L [28] | 84.9 | **82.3** | 87.5 | 18.0 |
| R50-DeAOT-L | **85.2** | 82.2 | **88.2** | **27.0** |
| SwB-AOT-L [28] | 85.4 | 82.4 | 88.4 | 12.1 |
| SwB-DeAOT-L | **86.2** | **83.1** | **89.2** | **15.4** |
| *Testing 2017 Split* | | | | |
| FEEL [22] | 57.8 | 55.2 | 60.5 | 1.9 |
| STM* [16] | 72.2 | 69.3 | 75.2 | - |
| CFBI [27] | 75.0 | 71.4 | 78.7 | 5.3 |
| CFBI* [27] | 76.6 | 73.0 | 80.1 | 2.9 |
| STCN [3] | 76.1 | 72.7 | 79.6 | 19.5⋆ |
| KMN* [19] | 77.2 | 74.1 | 80.3 | - |
| CFBI+* [29] | 78.0 | 74.4 | 81.6 | 3.4 |
| HMMN [20] | 78.6 | 74.7 | 82.5 | 3.4‡ |
| RPCM [26] | 79.2 | 75.8 | 82.6 | - |
| AOT-T [28] | 72.0 | 68.3 | 75.7 | 51.4 |
| DeAOT-T | **73.7** | **70.0** | **77.3** | **63.5** |
| AOT-S [28] | 73.9 | 70.3 | 77.5 | 40.0 |
| DeAOT-S | **75.4** | **71.9** | **79.0** | **49.2** |
| AOT-B [28] | 75.5 | 71.6 | 79.3 | 29.6 |
| DeAOT-B | **76.2** | **72.5** | **79.9** | **40.9** |
| AOT-L [28] | **78.3** | **74.3** | **82.3** | 18.7 |
| DeAOT-L | 77.9 | 74.1 | 81.7 | **28.5** |
| R50-AOT-L [28] | 79.6 | 75.9 | 83.3 | 18.0 |
| R50-DeAOT-L | **80.7** | **76.9** | **84.5** | **27.0** |
| SwB-AOT-L [28] | 81.2 | 77.3 | 85.1 | 12.1 |
| SwB-DeAOT-L | **82.8** | **78.9** | **86.7** | **15.4** |