# OpenReview forum: "Decoupling Features in Hierarchical Propagation for Video Object Segmentation"
_NeurIPS.cc/2022/Conference — NeurIPS 2022 Accept_

### Official Review · Reviewer_43Nu · 2022-07-08

**Rating:** 6
**Confidence:** 5
**Soundness:** 3 good
**Presentation:** 3 good
**Contribution:** 3 good

**Summary:**

This paper introduces a feature decoupling mechanism into the VOS task to solve the problem of visual information forgetting in the AOT method. Specifically, DeAOT decouples the hierarchical propagation of object-agnostic and object-specific embeddings by handling them in two independent branches. In addition, Gated Propagation Module with single-head attention is used to reduce the computational complexity. Experimental results on YouTube-VOS, DAVIS17, DAVIS16, and VOT20 demonstrate the performance gain of the proposed method.

**Questions:**

The incremental contributions. The main innovation is to expand the AOT model through a decoupling mechanism and gate attention. However, it is a common way to improve features' expression ability through decoupling, and gate attention is proposed in previous papers [23][27]. To prove the generalization ability of decouping mechnism, the authors should also combine it with other baseline method, such as STM.

Do this paper's main innovations (i.e., decoupling features) really play an essential role? In Table 3 (a), when the decoupling module is removed, the performance drops from 82.5 to 81.5, while when the GPM module is replaced, the performance drops more significant (82.5-> 80.3). This means GPM plays a more important role than decoupling feature.

For the head numbers in ablation experiment, the authors need to explain why the results of head numbers N_h=1 and N_h=8 are similar. Moreover, the performance of multiple-head is even inferior to that of single-head. This is conflict with the motivation of multiple-head mechanism. The authors may need to add other ablation results of N_h.

Typographical adjustments, Fig 4 goes too far from its text description.

 Minors: the abbreviation, i.e., AOT, should give the total words when first used.
grammar errors ‘Fig. 4 give qualitative comparisons’


**Limitations:**

Yes

**Strengths And Weaknesses:**

The in-depth analysis about the object-agnostic and object-specific gives a novel view for VOS task. In addition, two independent branches of the shared attention map do not significantly increase the model parameters.

The proposed gate attention reduces multiple-head attention to single-head attention, greatly reducing computational complexity while maintaining accuracy.

The paper is well-written and easy to read and understand.

---

> ### Author Response · Authors · 2022-08-02
> **Author Feedback**
>
> We appreciate you taking the time to share your comments and suggestions in the review assessment. We are glad that you and all the other reviewers recognize our contribution in decoupling object-agnostic and object-specific information, which helps DeAOT achieve state-of-the-art performance at impressive speed. All your comments are addressed point by point in the following.
>
> **Questions:**
>
> **Q1:** The incremental contributions.
>
> **A1:** The total contribution of DeAOT is not incremental but significant to the VOS field. The contribution of DeAOT can be separated into a major contribution and a minor contribution:
>
> (1)	The **major contribution** of DeAOT is based on the in-depth analysis of the object-agnostic and object-specific, which gives a novel view of the VOS task, as you mentioned. Based on the analysis, we carefully design a highly effective decoupling mechanism, which can generalize to different types (multi/single-object, long/short-term) of challenging benchmarks (YouTube-VOS 2018/2019, DAVIS 2017/2016, VOT 2020) and different AOT models with different backbones/attention layer numbers/inference types (CFBI-like or STM-like). We tried our best to ensure our DeAOT could generalize to all the possible VOS settings under a uniform framework. The characteristics of the used benchmarks and models are listed below:
>
> | Benchmark  | Characteristics |
> | - | - |
> | YouTube-VOS 2018/2019 | **large-scale, multi-object**, with **unseen categories**|
> |  DAVIS-2017 | small-scale, **multi-object** |
> |  DAVIS-2016  |  small-scale, **single-object**|
> |  VOT 2020  | small-scale, **single-object** with **long-term videos** (327 frames per video on average)|
>
> | Model  | Characteristics |
> | -| - |
> | DeAOT-T  |    **STM-like structure** with only one layer of matching and propagation. |
> |  DeAOT-S/B|    **Multiple layers** of matching and propagation. **CFBI-like inference**, considering only the reference frame as the long-term memory. |
> | (R50/SwinB-)DeAOT-L| **Multiple layers** of matching and propagation. **STM-like inference**, increasing the long-term memory per 5 frames.|
>
> (2)	The **minor contribution** is the gated attention/propagation for VOS. The design of such a module is responsible for compensating for the efficiency loss when utilizing the feature decoupling. In other words, the minor contribution serves the major contribution and is not independent of the major contribution. We agree with you that the gated attention/propagation is technically motivated by previous works [23][27]. However, the contribution of the gated propagation should not be independently assessed. Only by combining both the proposed feature decoupling and gated propagation, DeAOT achieves a comprehensive and significant improvement in performance and efficiency against previous VOS methods.
>
> In summary, the contribution of DeAOT is not incremental but significant. Besides, we have tried our best to conduct experiments on STM-like models and make all the models under a uniform framework. We believe our extensive experiments can demonstrate the generalization ability of the DeAOT framework.
>
> **Q2:** Do this paper's main innovations (i.e., decoupling features) really play an essential role?
>
> **A2:** Sorry for not getting you to understand our experimental results correctly. In Table 3 (a), replacing GPM with LSTT means removing the feature decoupling as well since the feature decoupling is a part of GPM in our default setting. When only the GPM module is replaced in AOT (without the feature decoupling), the performance drops from 81.5 to 80.3 (instead of from 82.5 to 80.3), according to Table 3 (a). Besides, according to our earlier experimental records, decoupling features in LSTT could bring a performance improvement of 1.2%, which is similar to the improvement of replacing LSTT with GPM (without the feature decoupling). In other words, both the feature decoupling and GPM are effective in improving VOS performance.
>
> **Q3:** Why the results of $N_h=1$ and $N_h=8$ are similar?
>
> **A3:** In the task of semi-supervised VOS, it is intuitively reasonable that single-head matching/propagation processes can achieve comparable or even better performance than multi-head processes. In the matching of objects in videos, it is impossible for each object to appear in multiple different locations in the same frame. In other words, the matching relationship of objects in different frames is a one-to-one correspondence, which can be represented by a single-head of attention map. Therefore, it is reasonable that single-head matching/propagation processes are capable of modeling multi-object matching and propagation. We will add the following discussion in the revision
>
> **Q4:** Typographical adjustments. The abbreviation, AOT. Grammar errors.
>
> **A4:** We sincerely thank you for your patience during the review, and we will correct all the typos in the revision.

---

> > ### Comment · Reviewer_43Nu · 2022-08-08
> > **rebuttal response**
> >
> > I have read the rebuttal carefully. I still have questions after reading the rebuttal.
> > 1. For Q2, the authors claim that both the feature decoupling and GPM are effective in improving VOS performance. However, the contribution of each component (feature decoupling and GPM ) is approximate. Thus, the main contribution, i.e., feature decoupling can not be regarded as the main innovations.
> > 2. For Q3, In the task of semi-supervised VOS, it is intuitively reasonable that single-head matching/propagation processes can achieve comparable or even better performance than multi-head processes. This statement is overclaim and questionable. The related proof or theoretical analysis should be provide.
> >
> > Overall, according to the current rebuttal, I change my score from Weak Accept to Borderline reject.

---

> > > ### Author Response · Authors · 2022-08-08
> > > **Author Feedback**
> > >
> > > Dear Reviewer 43Nu,
> > >
> > > We sincerely thank you for your time and your response. Your comments complete our paper and make it better.
> > >
> > > **We're sorry we couldn't answer your question in more detail earlier due to the character limit.** Regarding your new questions, our new responses are listed below:
> > >
> > > **New Q1**:
> > >
> > > For Q2, the authors claim that both the feature decoupling and GPM are effective in improving VOS performance. However, the contribution of each component (feature decoupling and GPM ) is approximate. Thus, the main contribution, i.e., feature decoupling can not be regarded as the main innovation.
> > >
> > > **A1:**
> > >
> > > The contribution of each component (feature decoupling and GPM) is not approximate, but only the performance gains are approximate. We consider feature decoupling as the major contribution since feature decoupling is a technically original and novel innovation in the field of semi-supervised VOS. We have a clear motivation and in-depth analysis of the feature decoupling for VOS. Compared to feature decoupling, GPM is technically motivated by some latest works [23][27]. Hence, the contribution of GPM is not comparable with feature decoupling since the contribution of GPM mostly comes from the hard design of the DeAOT framework.
> > >
> > > We tried our best to make DeAOT variants achieve the SOTA performance and efficiency across different benchmarks and model complexity. Regarding the current results, the DeAOT variants can still achieve the SOTA performance after losing half of the gains by removing GPM (but will lose some efficiency).
> > >
> > > **We hope the reviewer will not punish us because we have introduced the GPM module in order to improve DeAOT's efficiency. We just believe this will contribute to the VOS community further.**
> > >
> > > **New Q2:**
> > >
> > > For Q3, In the task of semi-supervised VOS, it is intuitively reasonable that single-head matching/propagation processes can achieve comparable or even better performance than multi-head processes. This statement is overclaiming and questionable. The related proof or theoretical analysis should be provided.
> > >
> > > **A2:**
> > >
> > > We didn't want to overclaim anything and were just trying to give a possible reason. In Semi-supervised VOS, most methods use single-head attention maps  (STM[32], KMN[40], SST[17], HMMN[41], STCN[11]) or matching maps (CFBI[57], CFBI+[59]) to match object patches from one-frame to another-frame. Intuitively, the optimal object matching between two frames is one-to-one matching for each object, since each object can not appear at multiple different places simultaneously. Based on such a motivation, KMN[40] proposed regularizing the matching/attention maps to be approximately one-to-one using the Gaussian kernel. Following such an innovation, MiVOS [a] proposed a stronger regularization method, the top-k memory strategy. The Gaussian kernel and the top-k memory strategy are utilized by STCN[11] as key tricks, significantly improving STCN's VOS performance (more than 2%).
> > >
> > > Even if we don't consider the characteristics of VOS, in another task, NLP, [23] and [27] **have found** that the presence of gating attention allows single-head attention performs better than multi-head attention. This motivated our minor contribution, and we designed the efficient GPM module for VOS.
> > >
> > > Based on the results, analyses, and findings from the above-mentioned works, we suppose it is reasonable that single-head matching/propagation processes can achieve comparable or even better performance than multi-head ones.
> > >
> > > We have to strengthen that the above analysis and conjecture are just based on our years of experience in the VOS field. Moreover, we are happy to discuss possible factors behind these open questions with experts in the field, and we respect all different opinions.
> > >
> > > **However, the use of a single head or multiple heads is only a small and empirical improvement of DeAOT. We hope the reviewer will not punish us since we shared some of our thoughts regarding this minor improvement.**
> > >
> > > If you have more questions, we will be happy to address them further.
> > >
> > > Best Regards,
> > >
> > > Authors
> > >
> > >
> > > [a] Modular Interactive Video Object Segmentation: Interaction-to-Mask, Propagation and Difference-Aware Fusion. CVPR 2021.

---

> > > > ### Comment · Reviewer_43Nu · 2022-08-09
> > > > **Response to A1**
> > > >
> > > > For A1,
> > > > the authors claim that  **We hope the reviewer will not punish us because we have introduced the GPM module in order to improve DeAOT's efficiency. We just believe this will contribute to the VOS community further.**
> > > >
> > > > Wow, firstly, I have never punished any ones. I just give my comments objectively.
> > > > Secondly, why do you just believe this will contribute to the VOS community further? Is this an overclaim? This work still follows the AOT [58] framework. The generalization ability of the proposed method has not been proved. Also, the performance promotion compared to AOT is moderate.

---

> > > > > ### Author Response · Authors · 2022-08-09
> > > > > **Author Feedback**
> > > > >
> > > > > Dear Reviewer 43Nu,
> > > > >
> > > > > Thanks for your valuable response again. Please forgive us if you felt offended. We didn't mean to offend you or anyone.
> > > > >
> > > > > We were trying to clear up your misunderstanding, and we bolded that sentence so that you could more clearly see our motivation for designing GPM.
> > > > >
> > > > > We are glad we can refine the questions step by step and solve them step by step during the communication process. Regarding your further questions, we list the responses below:
> > > > >
> > > > > **New Q1**:
> > > > >
> > > > > Why do you just believe this will contribute to the VOS community further?
> > > > >
> > > > > **A1**:
> > > > >
> > > > > In the process of our research on DeAOT, the successful introduction of feature decoupling allowed us to further raise the SOTA performance of the VOS field. While achieving breakthroughs in performance, feature decoupling would reduce algorithm efficiency.
> > > > >
> > > > > However, we hope that a good framework not only achieves breakthroughs in performance but also should reach a higher level of efficiency. We thought such a framework would make more sense for the VOS field. Hence, we further tried to design the GPM module to improve DeAOT's efficiency. Finally, we end up with improvements of VOS in both performance and efficiency.
> > > > >
> > > > > **New Q2**:
> > > > >
> > > > > The generalization ability of the proposed method has not been proved.
> > > > >
> > > > > **A2**:
> > > > >
> > > > > To prove the generalization ability of the proposed method, we have evaluated our method in four different architectures (including STM-like architectures) with three different backbones on four benchmarks, in which a new challenging benchmark (VOT2020) was used to further prove the generalization ability of our DeAOT.
> > > > >
> > > > > We tried our best to cover all popular settings of VOS in recent years. More details have been supplied in answer to the initial Q1. Please forgive us since we cannot list details here due to the character limitation.
> > > > >
> > > > > **New Q3**:
> > > > >
> > > > > The performance promotion compared to AOT is moderate.
> > > > >
> > > > > **A3**:
> > > > >
> > > > > The performance promotion of DeAOT is significant. For example, R50-DeAOT-L achieves a 1.9% improvement (from 84.1% to 86.0%) over R50-AOT-L on YouTube-VOS 2018, which is the most important benchmark of VOS. According to the progress of VOS in recent years, this improvement is not worse than the improvement made by previous SOTA methods in recent years, such as R50-AOT-L v.s. CFBI+ (1.3%, from 82.8% to 84.1%), CFBI+ v.s. CFBI (1.4%, from 81.4% to 82.8%), and CFBI v.s. STM (2.0%, from 79.4% to 81.4%).
> > > > >
> > > > > Considering it will be harder to improve performance on a stronger baseline, we suppose it is reasonable to claim that the performance improvement of DeAOT is significant.
> > > > >
> > > > > Thanks again for all your time and valuable comments. It seems that we have addressed most of your previous questions (if not, we are still happy to answer them for you), and we hope these new answers will address your new questions again.
> > > > >
> > > > > Best Regards,
> > > > >
> > > > > Authors

---

### Official Review · Reviewer_oq7a · 2022-07-10

**Rating:** 4
**Confidence:** 5
**Soundness:** 2 fair
**Presentation:** 2 fair
**Contribution:** 2 fair

**Summary:**

This paper proposes a hierarchical propagation module based on transformer which can keep the object-agnostic information in the videos. And it also proposes Gated Propagation Module for enhancing model efficiency. This method achieves high accuracy and efficiency.

**Questions:**

1.The performance of only containing ID branch is lacks in table 3 (c). For experimental integrity, the contribution of every module should be shown.
2.The author should list the module parameter and inference speed in table 3. The author needs to further demonstrate and verify whether the improvement in GPM is brought by adding more parameters.
3.The author needs to further analyze the effectiveness of modules in GP function. I want to know if gate (δ(U)) and DW-conv actually play an important role in this module
4.The authors mention that object-agnostic information fades in the video and use a single-backbone VOS model for hierarchical dual-branch propagation module validation. But the memory-based model such as STM takes a double backbone. Does this problem exist in the double backbone structure, and can the hierarchical dual-branch propagation module also improve the defects in the two-stage model?
5.I am interested in how the long-term and short-term memory images are divided and what the theoretical and experimental basis for this division.

**Limitations:**

In addition to the ablation experimental results and visualization results, the author also needs to analyze the effectiveness of the GPM and GP function from the theoretical basis and the feature visualization results.

**Strengths And Weaknesses:**

strengths:
The author alleviates the missing of object-agnostic information in deeper transformer modules by introducing hierarchical dual-branch propagation module. The GP function further improves the model efficiency while keeping high efficiency. This method has a certain originality.

Weakness:
The object-specific information can lead to the performance degradation of memory-based methods such as STM. STCN takes two object-agnostic features as Q and K to perform attention. This method extends this operation to a hierarchical propagation module and it is a continuation of this idea.

---

> ### Author Response · Authors · 2022-08-02
> **Author Feedback**
>
> We appreciate you taking the time to share your comments and suggestions in the review assessment. We are glad that you and all the other reviewers recognize our contribution in decoupling object-agnostic and object-specific information, which helps DeAOT achieve state-of-the-art performance at impressive speed. All your comments are addressed point by point in the following.
>
> **Weaknesses:**
>
> **Q:** DeAOT extends the operation of STCN.
>
> **A:** STCN is a great VOS job, but DeAOT extends the hierarchical propagation based on AOT instead of STCN. The AOT framework has two novel characteristics, i.e., the identification mechanism (responsible for matching, propagating, and decoding multiple objects together) and the hierarchical propagation (responsible for scaling VOS networks). These two characteristics are the essential difference between AOT and STM-like methods (e.g., STM and STCN). Figure 1 and Table 1 show that AOT has better performance, efficiency, and scalability than STCN.
>
> Although AOT shows strong results, our DeAOT still achieves a comprehensive and significant improvement in performance and efficiency over AOT. The total contribution of DeAOT is not incremental but significant to the VOS field:
>
> (1)   The **major contribution** of DeAOT is the feature decoupling method, which can generalize to different types (multi/single-object, long/short-term) of challenging benchmarks and different VOS models with different backbones/propagation layer numbers/inference types (CFBI-like or STM-like). We tried our best to ensure our DeAOT could generalize to all the possible VOS settings under a uniform framework.
>
> (2)   The **minor contribution** is the gated propagation responsible for improving DeAOT's efficiency. Combining both feature decoupling and gated propagation, DeAOT achieves a significant improvement in performance and efficiency against previous VOS methods.
>
> **Questions:**
>
> **Q1:** Only containing ID branch is lacking.
>
> **A1:** Thanks for your suggestion. We have considered the experiment of using only the ID branch, but we found it is not applicable. When predicting the current frame mask, the initial information (which can be used to match objects) is only the visual information given by the current frame image. In other words, it is necessary and inevitable to use visual embeddings to compute matching/attention maps.
>
> **Q2:** Whether the improvement is brought by adding more parameters.
>
> **A2:** Thanks for your suggestion, and we will supply the information of parameters in the revision. Here, we show some important parameter comparisons below (all the models use ResNet-50 as the backbone):
>
> | Model | STM | HMMN | STCN | R50-AOT-L | R50-DeAOT-L |
> | ----- | --- | ---- | ---- | --------- | ----------- |
> | YouTube-VOS accuracy   |  79.4   |   82.6   |   83.0   |     84.1      |    **86.0**         |
> | fps   |  -   |   -   |   8.4   |    14.9   |    **22.4**     |
> | Param (M)      |  34.0   |  42.8    |   54.4   |    **14.9**       |        19.8     |
>
> As shown, R50-DeAOT-L significantly outperforms STCN and R50-AOT-L on accuracy and run-time speed. However, R50-DeAOT-L introduces only 5M parameters (which is inevitable due to feature decoupling) compared to R50-AOT-L and uses only 36% of the parameters of STCN (19.8M v.s. 54.4M). Besides, the careful design of GPM makes DeAOT run even much faster than AOT.
>
> **Q3:** The role of gate (δ(U)) and DW-Conv.
>
> **A3:** According to our experimental records, removing δ(U)/DW-Conv from DeAOT-S degraded the performance by 1.9%/2.7%. δ(U) is necessary for improving the ability of non-linear transformation of GPM. Without δ(U), the non-linear ability of GPM only comes from the LN modules and is not sufficient to fit complex functions. The DW-Conv is critical in enlarging the receptive fields of DeAOT and AOT (AOT uses a DW-Conv layer in each feed-forward module). Without DW-Conv, the performance of DeAOT/AOT will degrade by about 2%~3%.
>
> **Q4:** Does the problem of fading object-agnostic information exist in STM-like double backbone structure?
>
> **A4:** During the design of DeAOT, we also considered using STM-like double backbones. However, we didn't find any obvious performance gains by replacing the ID bank of DeAOT/AOT with a ResNet-18/34 backbone. In this regard, our analysis is that the information contained in the mask is not as rich as the image, and a simple ID bank can generate the mask embedding well.
>
> **Q5:** How to divide the long-term and short-term memories?
>
> **A5:** The long short-term memories are introduced in AOT and not our contribution. Their theoretical and experimental basis can be found in the original AOT paper.
>
> **Q6:** The theoretical basis of GPM.
>
> **A6:** Thanks for your suggestion. Although the GPM is only a minor contribution of DeAOT, we are glad to study the theoretical basis of GPM in future works. If possible, we will add relevant progress in the revision.

---

### Official Review · Reviewer_vqMg · 2022-07-11

**Rating:** 7
**Confidence:** 4
**Soundness:** 4 excellent
**Presentation:** 4 excellent
**Contribution:** 3 good

**Summary:**

The paper tackles the problem of video object segmentation by extending the Associating Objects with Transformers (AOT) approach. AOT uses multiple Transformer blocks to gradually integrate target information from the previous frame and masks into the current frame features. The encoded features are then decoder to obtain the target mask. The deeper Transformer blocks in AOT operate on features which are target-specific, i.e. which contain encoding of the target mask information. The authors claim that using such features to compute the attention maps inside Transformer harms performance since they contain less target agnostic visual information. To address this, the authors propose to decouple the propagation of target-specific and target-agnostic features using two parallel transformer branches. Both the branches share attention maps, computed using the target-agnostic features. Additionally, the authors propose an efficient Gated Propagation Module (GPM) instead of Long-Short term Transformer (LSTT) used in AOT to propagate the features. GPM uses single-head attention, as opposed to multi-headed attention in LSTT. Additionally it uses a gating function to modulate the propagated features. The use of GPM compensates for the extra computation incurred due to use of two transformer branches. Overall, the proposed methods consistently outperforms baseline AOT on 4 benchmarks, across 3 different backbone networks (+1.6% on YouTube-VOS 2019), obtaining state-of-the-art results on these benchmarks. The proposed method is also faster than AOT.

**Questions:**

Please see weaknesses.

**Limitations:**

The authors discuss the limitations, albeit in the supplementary material.

**Strengths And Weaknesses:**

## Strengths

**S1**: The proposed approach obtains state-of-the-art results on YouTube-VOS, DAVIS, and VOT2020 benchmarks while operating at impressive speed (>10 FPS).

**S2**: The proposed approach generally outperforms the baseline AOT across YouTube-VOS, DAVIS, and VOT2020 benchmarks, when using 5 different backbones/configurations. Furthermore the proposed approach is faster then AOT.

**S3**: The paper is well written, and the authors clearly introduce the issue with the baseline AOT. This claim is further experimentally validated (Figure 2, Table 3a). The proposed approach is shown to mitigate this issue.

**S4**: The proposed Gated Propagation Module (GPM) is interesting and seems to provide a clear improvement over LSTT block, while being faster (Table 3a, 3b).

**S5**: The authors perform a thorough analysis of the proposed approach, showing the impact of each component.

## Weaknesses
I do not have any major issues with the paper. Some minor comments to be addresses

**W1**:  It wasn't clear to me how the gating embedding U is obtained. The authors should clarify this.

**W2**: It would be interesting to see the performance of LSTT with decoupling features (i.e. an extra entry in Table 3a for completeness).

**W3**: Are both the encoded visual features as well as the ID features used to obtain the target mask, or only the ID features?

**W4**: The current approach performs a strict separation of object-specific and object-agnostic features, and mainly uses the object-agnostic features to compute the attention masks. However I imagine that certain object-specific information could also be helpful when computing attention masks. Thus would it make sense to instead have two branches, one which is used to compute the attention masks, and the second used to obtain the final mask. Both of these branch can take both the image and object masks as input, and determine whether to use the ID information or not during the training process.

---

> ### Author Response · Authors · 2022-08-02
> **Author Feedback**
>
> We appreciate you taking the time to share your comments and suggestions in the review assessment. We are glad that you and all the other reviewers recognize our contribution in decoupling object-agnostic and object-specific information, which helps DeAOT achieve state-of-the-art performance at impressive speed. All your comments are addressed point by point in the following.
>
> **Questions:**
>
> **Q1:** It wasn't clear to me how the gating embedding U is obtained. The authors should clarify this.
>
> **A1:** Sorry for not being able to give you a simple understanding of the meaning of the gating embedding U, and we will add more descriptions in the revision to help the readers understand it. In GPM, the way to obtain the gating embedding U is simple and similar to the way to obtain other commonly-used embeddings (e.g., Q, K, V). In attention mechanisms, Q, K, and V are obtained after linearly projecting a feature (or three different features) by their weights, $W^{Q}, W^{K}, W^{V}$ (notably, we set $W^{Q} = W^{K}$ in GPM to share the embedding space of Q and K). In GPM, the gating embedding U is also obtained after projecting a feature by a learnable weight, $W^{U}$. In the object-agnostic/specific branch of GPM, the projected feature is the visual/mask feature (I/M). $W^{U}$ is not shared among different GPM modules and different branches. More details can be found in Eq. 6, 7, 10, and 11.
>
> **Q2:** It would be interesting to see the performance of LSTT with decoupling features (i.e., an extra entry in Table 3a for completeness).
>
> **A2:** Thanks for your suggestion, and we will supply the related results in the revision. According to our earlier experimental records, decoupling features in LSTT (putting Long/Short Term Attn before Self-Attn) could bring a performance improvement of 1.2% (J&F, YouTube-VOS) to AOT-S, which is consistent with the experimental phenomena of GPM in Table 3(a). However, decoupling features in LSTT reduced run-time speed by more than 5 fps. The main reason is that the doubled multi-head propagations are computationally intensive.
>
> **Q3:** Are both the encoded visual features as well as the ID features used to obtain the target mask or only the ID features?
>
> **A3:** To predict the target mask, both the visual (object-agnostic) and ID (object-specific) features are concatenated and forwarded to the decoder network. We also tried to use only the ID features to decode the mask prediction but didn't find a noticeable improvement in performance or efficiency (the decoder network, FPN, is light-weight).
>
> **Q4:** The current approach performs a strict separation of object-specific and object-agnostic features and mainly uses the object-agnostic features to compute the attention masks. However, I imagine that certain object-specific information could also be helpful when computing attention masks. Thus would it make sense to instead have two branches, one which is used to compute the attention masks and the second used to obtain the final mask. Both of these branches can take both the image and object masks as input and determine whether to use the ID information or not during the training process.
>
> **A4:** Thanks for your detailed suggestion, which is really worth thinking about in future works. The success of DeAOT feature decoupling indicates that it is reasonable and effective to use one branch to compute attention masks and another branch to obtain the final mask (as you mentioned). But the current experimental results in Table 3(c) show that using only visual embeddings to compute attention masks leads to better performance. A possible reason is that using both visual and mask embeddings may make the networks overfitted to the training objects' shape patterns. In summary, how to effectively use both the image and object masks in computing attention masks is still an open problem and requires future studies.

---

> > ### Comment · Reviewer_vqMg · 2022-08-07
> > **Thank you for the reponse**
> >
> > Thanks for the detailed response to my questions. I will stick with my original rating and recommend acceptance.

---

> > > ### Author Response · Authors · 2022-08-08
> > > **Thanks for your valuable comments**
> > >
> > > Dear Reviewer vqMg,
> > >
> > > We sincerely thank you for your valuable comments during the review procedure, and we are glad you recognize our DeAOT's novelty and contributions to the VOS field!
> > >
> > > Best Regards,
> > >
> > > Authors

---

### Official Review · Reviewer_Kx9L · 2022-07-12

**Rating:** 7
**Confidence:** 4
**Soundness:** 3 good
**Presentation:** 3 good
**Contribution:** 3 good

**Summary:**

This paper proposed an improved AOT method called DeAOT. The main idea of DeAOT is to separate representation learning from identity information. This means using two GPMs instead of the original LSTT in AOT. The Gated Propagation Module(GPM) is faster than LSTT and achieves better performance.

**Questions:**

Compared to AOT, the difference between DeAOT seems to be only in the design of Propagation Module, and specifically the difference between LSTT and GPM. Therefore, it is more important to explain the more subtle differences between LSTT and GPM and to explore what really works in the design of the GPM. For example:
- the effect of removing FFN, or whether introducing FFN would result in better speed-performance tradeoff
- the impact of swapping Self-Attn(Prop) and Long/Short Term Attn(Prop)
- the difference between sharing an attention map or not

From the ablation information provided in the current experiments(Table 3), we could find that GPM still outperforms LSTT even without disentangle representation learning and id information, so it is more important to analyze the role of GPM clearly. From another perspective, GPM is the core of this paper and it would be detrimental to the quality of this paper if the mechanism of GPM could not be well explained.

**Limitations:**

No but not need.

**Strengths And Weaknesses:**

Strengths
- Great performance on several VOS/VOT benchmarks.
- Intuitive and effective design about disentangle the representation learning and id information.
- GPM balances the inference speed and performance.

Weaknesses
- Lack of ablation studies on GPM.

---

> ### Author Response · Authors · 2022-08-02
> **Author Feedback**
>
> We appreciate you taking the time to share your comments and suggestions in the review assessment. We are glad that you and all the other reviewers recognize our contribution in decoupling object-agnostic and object-specific information, which helps DeAOT achieve state-of-the-art performance at impressive speed. All your comments are addressed point by point in the following.
>
>
> **Questions:**
>
> **Q1:** The effect of removing FFN, or whether introducing FFN would result in a better speed-performance tradeoff.
>
> **A1:** Removing FFN from the LSTT of AOT-S will significantly degrade its performance by more than 3% (J&F, YouTube-VOS) but only improve run-time speed by about 2 fps. Such a tradeoff is worthless. Without FFN, the non-linear transformations within LSTT only come from the LN modules and are insufficient to fit complex functions. By contrast, in GPM, the GP function compensates for the lack of non-linear transformations by introducing the gating embedding U, which is followed by a non-linear activation (SiLU/Swish in our setting).
>
>
> **Q2:** The impact of swapping Self-Attn (Prop) and Long/Short Term Attn (Prop).
>
> **A2:** Putting Long/Short Term Attn(Prop) before Self-Attn(Prop) is not a trick for DeAOT but is an inevitable design. The Self-Prop takes two input branches, i.e., object-agnostic visual embedding and object-specific mask embedding. Suppose we put Self-Prop before Long/Short Term Prop. In that case, there will be no object-specific mask embedding for the first Self-Prop module since the first module in GPM can only take the encoder network’s output as input, which is an object-agnostic visual embedding. But if we put Self-Prop after Long/Short Term Prop, the first Long/Short Term Prop module can generate an object-specific embedding by propagating mask embeddings from past frames to the current frame. Thus, the next Self-Prop module can have two branches of input.
>
> Although putting Long/Short Term Attn(Prop) before Self-Attn(Prop) is necessary for DeAOT, we were also curious about its effect on the LSTT of AOT. According to our experimental records, putting Long/Short Term Attn before Self-Attn positively affected AOT-T (about 0.4% improvement, J&F, YouTube-VOS) but showed no obvious improvement in deeper networks (AOT-S/AOT-B/AOT-L/R50-AOT-L). Overall, the swapping will not make a significant performance difference.
>
>
> **Q3:** The difference between sharing an attention map or not.
>
> **A3:** According to our earlier experimental records, decoupling features in LSTT (putting Long/Short Term Attn before Self-Attn) could bring a performance improvement of 1.2% (J&F, YouTube-VOS) to AOT-S, which is consistent with the experimental phenomena of GPM in Table 3(a). However, decoupling features in LSTT reduced run-time speed by more than 5 fps. The main reason is that the doubled multi-head propagations are computationally intensive.
>
> Regarding the discussion of different possible ways to share attention maps, we have supplied detailed ablation studies in Table 3 (c). Moreover, please refer to Table 3 (a) for the GPM without sharing attention maps.
>
>
> **Weaknesses:**
>
> **Q:** Lack of ablation studies on GPM. It is more important to explain the more subtle differences between LSTT and GPM.
>
> **A:** Thanks for the suggestion. In the above answers, we addressed all your questions about the subtle differences between LSTT and GPM. In addition, we will add all these valuable discussions into the revised version or the supplementary material.

---

> > ### Comment · Reviewer_Kx9L · 2022-08-08
> > **Reply to the authors**
> >
> > Thanks for the authors' response and I think my concerns have been adequately addressed. I will raise my score to a certain accept.

---

### Comment · Area_Chair_PiyB · 2022-08-06
**Discussion**


Dear Reviewers,

Thank you for reviewing this paper. Since authors have submitted their rebuttal, please check if the rebuttal addresses your concerns. Please also check the comments from other reviewers, and if you have any question, please discuss with authors in OpenReview soon.

Best Regards,

AC

---

### Meta-Review · Area_Chair_PiyB · 2022-08-24

**Recommendation:** Accept
**Confidence:** Certain

**Metareview:**

The paper obtains three accept and one borderline reject recommendations. Yet all reviewers pointed out that the paper has novelty and originality in the domain of video object segmentation, and also the method works quite well on the tested datasets. The reviewer recommending rejection does not comment at the post-rebuttal phase, and the AC has checked authors' response, and most concerns of the reviewer have been addressed - though the theoretical analysis can be a future work, the feature visualization still can be done in the camera ready version. Authors also need to carefully prepare the camera ready version, since all the reviewers indeed give valuable comments, which are helpful for the paper.

**Award:**

No

---

### Decision · Program_Chairs · 2022-09-14

Accept